# A Comparison of Emergency Department Revisit Rates of Pediatric Patients between Pre-COVID-19 and COVID-19 Periods

**DOI:** 10.3390/children9071003

**Published:** 2022-07-04

**Authors:** Myeong Namgung, Dong Hoon Lee, Sung Jin Bae, Ho Sub Chung, Ji Young Park, Keon Kim, Choung Ah Lee, Duk Ho Kim, Eui Chung Kim, Jee Yong Lim, Sang Soo Han, Yoon Hee Choi

**Affiliations:** 1Department of Emergency Medicine, College of Medicine, Chung-Ang University, Seoul 06973, Korea; myeong15180@caumc.or.kr; 2Department of Emergency Medicine, College of Medicine, Chung-Ang University Gwangmyeong Hospital, Chung-Ang University, Gwangmyeong-si 14353, Korea; uzimuz85@gmail.com (S.J.B.); hoshap@cau.ac.kr (H.S.C.); 3Department of Pediatrics, College of Medicine, Chung-Ang University, Seoul 06973, Korea; jypark@caumc.or.kr; 4Department of Emergency Medicine, College of Medicine, Ewha Womans University, Seoul 07804, Korea; mikky5163@gmail.com; 5Department of Emergency Medicine, Hallym University Dongtan Sacred Heart Hospital, Hwaseong-si 18450, Korea; cuccum@hanmail.net; 6Department of Emergency Medicine, Eulji University, Seoul 01830, Korea; ejrghsla1@naver.com; 7CHA Bundang Medical Center, Department of Emergency Medicine, Seongnam-si 13496, Korea; oct01@cha.ac.kr; 8Department of Emergency Medicine, Seoul St. Mary’s Hospital, Seoul 06591, Korea; ny1117@catholic.ac.kr; 9Department of Emergency Medicine, Soonchunhyang University Bucheon Hospital, Bucheon-si 14584, Korea; brayden0819@daum.net; 10Department of Emergency Medicine, College of Medicine, Ewha Womans University Mokdong Hospital, Seoul 07985, Korea; unii@ewha.ac.kr

**Keywords:** revisit to emergency department, COVID-19, pediatric patients

## Abstract

Unscheduled revisits to emergency departments (EDs) are important because they indicate the quality of emergency care. However, the characteristics of pediatric patients visiting EDs changed during the coronavirus disease (COVID-19) pandemic, and these changes may have affected their revisit patterns. Therefore, we aimed to compare the ED revisit patterns of pediatric patients between the pre-COVID-19 and COVID-19 periods. This retrospective multicenter study included patients aged below 18 years who visited the ED in the pre-COVID-19 and COVID-19 periods. ED revisit rates were analyzed using five age groups and three visit-revisit intervals. In the pre-COVID-19 period, the revisit rates decreased with increasing age. In the COVID-19 period, the revisit rates were the lowest for the group aged 4–6 years, and the rates increased for those aged ≥7 years. In conclusion, there were changes in the patterns of revisit rates of pediatric patients according to age between the pre-COVID-19 and COVID-19 periods. Therefore, it is necessary to identify the reasons for revisits according to age and establish strategies to reduce the revisit rates of pediatric patients.

## 1. Introduction

An unscheduled revisit to emergency departments (EDs) is usually considered a quality indicator of emergency care because it is associated with an inadequate initial evaluation, treatment, and follow-up instructions [1,2,3,4]. The risks of hospitalization and mortality are higher at the revisit than at the first ED visit [5,6,7]. Additionally, unscheduled ED revisits increase health care costs [8] and can contribute to overcrowding in EDs, especially if the revisits are unnecessary [9]. Therefore, it is important to identify the factors causing unscheduled ED revisits of patients and develop strategies to minimize them.

A few studies on ED revisits were conducted on pediatric patients. In previous studies, the ED revisit rates of pediatric patients varied from 2.2% to 8.0% [10,11,12,13,14,15,16,17,18], and the number of pediatric revisits to EDs was reported to increase each year [19]. The ED revisit rates were high among infants, toddlers [9,12,14,17,18,19], and children with chronic diseases [11,20]. The hospitalization rate of pediatric patients was higher at the revisit than at their first visit, similar to the pattern reported in adults [13]. Furthermore, few previous studies showed the factors contributing to revisits, such as diagnoses and symptoms. The results of two studies that analyzed the revisit rates according to diagnosis were not applicable to all the pediatric patients, as one of those studies included pediatric patients with chronic disease [9] and the other included neonates [21].

Moreover, because of the coronavirus disease (COVID-19) pandemic, the characteristics of pediatric patients visiting the ED and the clinical process of emergent care have changed. During the COVID-19 period, the number of pediatric patients visiting the ED decreased. Particularly patients with respiratory and infectious diseases and low acuity had a short length of stay (LOS) in the ED [22,23], while those with fever had a prolonged stay [24]. Furthermore, the management strategies for pediatric patients with trauma also changed [25]. Thus, we hypothesized that these changes would impact the characteristics of pediatric ED revisits.

Therefore, we aimed to compare the characteristics of pediatric patients who revisited the ED in pre-COVID-19 and COVID-19 periods. We primarily analyzed the ED revisit rates according to age and visit-revisit interval in each period. The secondary objective was to identify the diagnosis and disposition of pediatric patients who revisited the EDs in the pre-COVID-19 and COVID-19 periods.

## 2. Materials and Methods

### 2.1. Study Design and Period

This retrospective multicenter study included pediatric patients aged below 18 years who visited the ED before and after the COVID-19 pandemic in seven tertiary hospitals located in the metropolitan area of South Korea. The annual number of pediatric patients who visited the EDs in these hospitals ranged from 8000 to 36,000.

The “COVID-19 period” was defined as the period from March 2020 to February 2021 because after the first case of COVID-19 was confirmed on 20 January 2020, in South Korea, the number of patients gradually increased leading to the full-blown COVID-19 pandemic in March 2020. The “pre-COVID-19 period” was defined as the period from March 2018 to February 2019.

### 2.2. Study Population

Patients aged <18 years, who visited ED during the study period, were included in this study. We excluded the first visits made for (1) non-medical reasons; first visits in which (2) the patient left against medical advice, (3) died in the ED, (4) was admitted to the general ward or intensive care unit (ICU), or (5) was transferred to another facility; and first visits in which (6) the patient’s disposition was unknown. The visits that remained after the application of these exclusion criteria were defined as index visits. Of these, only the revisits for medical reasons within 7 days of the first visit were included in our analysis.

### 2.3. Identification of Index Visit and Revisit 

All ED visits during the study period were classified either as an index visit or revisit. “Index visit” was defined as the first ED visit among consecutive ED visits, and “revisit” was defined as an ED visit immediately after the index visit. Therefore, the same patient could have multiple index visits and revisits in the current analysis, and a revisit could be the index visit of the next revisit. Because we only used the data extracted from the National Emergency Department Information System (NEDIS) using the registration number of patients and did not analyze medical records, we were unable to confirm whether the patients revisited the ED for the same diagnosis/symptoms as those in their index visits. Thus, we could not confirm whether the diagnosis at the revisit was the same as that at the index visit in this analysis. Similarly, we could not confirm the purpose of the revisit and whether it was scheduled or unscheduled.

### 2.4. Calculation of the Revisit Rate

The revisit rate was calculated as the ratio of the number of revisits to the number of index visits. To identify the differences in the revisit rates as per the age and visit-revisit interval, we divided the patients into five age groups and three visit-revisit interval groups. The patients were divided into the following age groups: (1) aged below 1 year (infants), aged 1–3 years (toddlers), aged 4–6 years (preschoolers), aged 7–12 years (elementary school students), and aged 13–17 years (middle and high school students). The visit-revisit interval was classified as follows: revisit to the ED within 1, 3, and 7 days. Most studies on ED revisits of pediatric patients use a visit-revisit interval of 3 to 7 days [13,16,26]. However, we also included revisits within 1 day because, according to Perry et al., ED revisits within 24 h could represent serious deficiencies in emergency health care [21].

### 2.5. Data Collection and Outcome Measurement

We obtained data from the NEDIS database, which is a national database system that transmits and analyzes the information of patients visiting the ED in real time. The following data were collected: age, sex, Korea Triage and Acuity Scale ((KTAS) resuscitation, level I; emergency, level II; urgent, level III; less urgent, level IV; and nonurgent, level V), transportation to the hospital, chief complaint, vital signs upon arrival at the ED, time variables (visit, discharge, and admission), primary diagnosis, and disposition (discharge, admission, transfer, and death)

The primary outcome was the ED revisit rates of each age group within 1, 3, and 7 days in the pre-COVID-19 and COVID-19 periods. The secondary outcomes were the final diagnosis and disposition at the revisit in both periods. The final diagnoses at the revisits were classified according to the International Statistical Classification of Diseases and Related Health Problems-10 (ICD-10). 

### 2.6. Statistical Analysis

All statistical analyses were performed using SPSS version 26.0 (IBM, Chicago, IL, USA). Categorical data were expressed as numbers and percentages. Continuous data were expressed as mean with standard deviation if they showed normal distribution or as median with interquartile range if they showed non-normal distribution. Chi-squared test was performed for categorical variables, and an analysis of variance was used for continuous variables. A *p*-value < 0.05 was considered statistically significant. 

### 2.7. Ethical Consideration

This study was approved by the institutional review boards (IRB No. 2111-067-19396). This retrospective study used the hospital database, and all patients were anonymized before the study; therefore, the need for informed consent was waived by the institutional review boards.

## 3. Results

### 3.1. Revisit Rates in Pre-COVID-19 and COVID-19 Periods

Figure 1A,B present flowcharts of ED visits recorded during the study period. In total, there were 464,692 ED visits in the pre-COVID-19 period and 317,256 ED visits in the COVID-19 period; among these visits, 134,901 and 54,725 were made by pediatric patients during each period, respectively. After excluding the patients according to the exclusion criteria, the remaining 71,564 and 23,994 visits were used as index visits in the pre-COVID-19 and COVID-19 periods, respectively. Finally, 5035 revisits in the pre-COVID-19 period and 1418 revisits in the COVID-19 period within 7 days were included in the analysis. Among the 71,564 index visits during the pre-COVID-19 period, there were 2924 (4.1%), 4304 (6.0%), and 5035 (7.0%) revisits within 1, 3, and 7 days, respectively (Figure 1A). Furthermore, among the 23,994 index visits during the COVID-19 period, there were 865 (3.6%), 1223 (5.1%), and 1418 (5.9%) revisits within 1, 3, and 7 days, respectively (Figure 1B).

Figure 2 shows the ED revisit rates according to age within 1, 3, and 7 days. In the pre-COVID-19 period, the revisit rates for all time intervals decreased with increasing age. The revisit rates of children aged <1 year were the highest for all intervals at 6.2 %, 9.3 %, and 10.8 %, respectively, and those of children aged 13–17 years were the lowest at 1.8%, 2.7%, and 3.3%, respectively. In the COVID-19 period, the revisit rates of children aged <1 year were the highest at 4.3%, 6.4%, and 7.6%, respectively, while the revisit rates of those aged 4–6 years were the lowest at 2.5%, 3.9%, and 4.6%, respectively; furthermore, the revisit rates increased among those aged ≥7 years. The gap in revisit rates of children aged <1 year and those aged 1–3 years during the COVID-19 period had narrowed compared to that in the pre-COVID-19 period. In particular, in the COVID-19 period, the revisit rate within 1 day of children aged <1 year was 4.3%, slightly lower than the revisit rate of 4.5% of those aged 1–3 years.

### 3.2. Baseline Characteristics

The baseline characteristics of the pediatric patients at the index visit and at the 7th-day revisit are detailed in Table 1. The average age at index visits was higher than that at revisits in both the study periods (in the pre-COVID-19 period: age at index visit, 4.9 ± 4.7 years, age at revisit, 3.8 ± 3.9 years, *p* < 0.001; in the COVID-19 period: age at index visit, 5.7 ± 5.3 years, age at revisit, 4.9 ± 5.3 years, *p* < 0.001). In the pre-COVID-19 period, the proportions of patients with KTAS levels 2 and 3 (0.7% and 40.2%, respectively) were lower at the revisits than at the index visits (3.1% and 56.3%, respectively) (*p* < 0.001). However, in the COVID-19 period, the proportions of patients with KTAS levels 2 and 3 (4.8% and 63.4%) were higher at the revisits than at the index visits (3.6% and 53.4%) (*p* < 0.001). The mean LOS was longer at revisit than at index visit during both the study periods. (105 ± 96 min vs. 113 ± 67 min in the pre-COVID-19 period, *p* < 0.001; 107 ± 102 min vs. 169 ± 189 min in the COVID-19 period, *p* < 0.001).

### 3.3. Reduction in the Index Visit and Revisit Rates

In order to understand the reason behind changes in the revisit rates (Figure 2), we analyzed the reduction in the index visit and revisit rates in each study period. The reduction rate of index visits and revisits was calculated as the ratio of the difference in the number of index visits and revisits within 7 days between the pre-COVID-19 and COVID-19 period divided by the number of index visits and revisits in the pre-COVID-19 period, respectively. 

The total number of index visits and revisits in both periods and the calculated reduction rates are shown in Figure 3. Although the total number of index visits and revisits for all age groups decreased in the COVID-19 period, the reduction rates between the index visits and revisits varied according to age. In the COVID-19 period, in the 0–6 years group, the reduction rate of index visits was higher than that of revisits, while in the 7–17 years group, the reduction rate of the index visits was lower than that of revisits.

### 3.4. Diagnosis at Revisit According to Age

Table 2 shows the three most common diagnoses during the pre-COVID-19 and COVID-19 periods for each age group. In both of the study periods, the most common diagnosis was that of infectious and parasitic diseases for all age groups, except the 1–3 years group, and most revisits occurred within 3 and 7 days. In the pre-COVID-19 period, the top three diagnoses in the 7–12 years group were infectious and parasitic diseases, respiratory disease, and digestive disease. However, during the COVID-19 period, the top three diagnoses were infectious and parasitic diseases, digestive diseases, and skin and subcutaneous tissue disease. In the 13–17 years group, the third most common diagnosis at revisit within 7 days was respiratory disease in the pre-COVID-19 period and mental and behavioral disorders in the COVID-19 period.

### 3.5. Disposition

For all age groups, except the 13–17 years group, the rate of admission to the general ward was higher in the COVID-19 period than in the pre-COVID-19 period. The rate of admission to the ICU was higher in the COVID-19 period than in the pre-COVID-19 period only for the <1 year group within 1, 3, and 7 days and for the 13–17 years group within 3 and 7 days (Table 3).

## 4. Discussion

To the best of our knowledge, this is the first report of comparison between the revisit rates of pediatric patients to the ED between the pre-COVID-19 and COVID-19 period. We found that the ED revisit rates of pediatric patients in the COVID-19 period were different from those in the pre-COVID-19 period. In the pre-COVID-19 period, the revisit rates decreased with increasing age. However, in the COVID-19 period, the revisit rates were the lowest for the 4–6 years group and increased among children aged ≥7 years. This finding is specific for pediatric patients because there was no change in the pattern of ED revisit rates among adults between the pre-COVID-19 and COVID-19 periods. For adults, the revisit rates increased with advancing age during both periods (Appendix A).

We further analyzed the reduction rates of index visits and revisits. As shown in Figure 3, for those aged 0–6 years, the reduction rate of index visits was lower than that of revisits, and for those aged 7–17 years, the reduction rate of index visits was higher than that of revisits. The revisit rate was calculated as the ratio of the number of index visits to the number of revisits. Therefore, the revisit rates increased for those aged 7–17 years in the COVID-19 period because the decrease in the number of revisits was lower than the decrease in the number of index visits. On the contrary, for those aged 0–6 years, as the decrease in the number of index visits was greater than that in the number of revisits, the revisit rate during the COVID-19 period decreased. Although further studies are needed to explain the reasons for the changes in the revisit rates in the COVID-19 period, to some extent, these changes can be explained by the changes in the ratio of index visits to revisits according to age.

Compared to the pre-COVID-19 period, in the COVID-19 period, there were changes in the diagnoses and dispositions of pediatric patients who revisited the ED. In the COVID-19 period, the proportion of patients who revisited the ED with respiratory diseases had decreased due to a decrease in the total number of patients with respiratory diseases [22,23]. The measures to prevent the spread of COVID-19 infection, including social distancing measures, hand washing, wearing a mask, and school closures, helped prevent the transmission of respiratory diseases [22]. Furthermore, because patients suspected of having the COVID-19 infection, such as those with fever or respiratory symptoms, used isolation rooms during the COVID-19 period, the number of patients that could be accommodated in the ED was inevitably reduced [24]. On the other hand, the proportion of admissions to the general ward decreased in the COVID-19 period for all age groups, except for the 13–17 years group; this finding may be associated with the decrease in the number of patients with respiratory diseases, considering that patients with respiratory complaints reportedly have a high risk of admission on revisit according to previous studies [14,17].

Another change observed in the COVID-19 period was that the third most common diagnosis in the 13–17 years group at a revisit to the ED within 7 days was mental and behavioral disorders. This means that adolescents experienced psychological problems during the pandemic. According to Panda et al., children were primarily affected by the pandemic and quarantine and experienced psychological problems including anxiety, depression, irritability, boredom, inattention, and fear of COVID-19 [27]. Another study also showed that the rates of ED visits of adolescents with suicide attempts significantly increased during the COVID-19 period [28]. We thought that the increase in the number of adolescent patients with mental health problems might be related to an increase in the revisit rate of adolescents in the COVID-19 period, given that the revisit rate was as high as 45% [29]. Therefore, in order to reduce the revisit rate, emergency physicians or psychiatrists need to pay attention to the mental health of adolescents.

This study has several limitations. First, the data were extracted only from the NEDIS database using the registration number of patients; hence, we were unable to confirm whether the patients revisited the ED for the same diagnosis/symptoms as those reported in their index visits. Therefore, we could have included patients who revisited the ED with a different diagnosis or for different symptoms from those reported at the index visit. To overcome this, we only included patients who visited the ED for medical reasons except for trauma. Second, because this was a retrospective study, we could not confirm the purpose of revisit and whether it was a scheduled or unscheduled visit. Finally, we only analyzed patients who revisited the same hospital as the index visit. Thus, the revisit rates in this study could be underestimated.

## 5. Conclusions

In summary, there was a change in the revisit rates of pediatric patients according to age between the pre-COVID-19 and COVID-19 periods. The revisit rates of children aged 0–6 years were lower, and the revisit rates of children aged 7–17 years were higher in the COVID-19 period than those in the pre-COVID-19 period. These findings suggest that individualized interventions and strategies according to age are needed to reduce the revisits of pediatric patients to the ED. Therefore, further studies are needed to explain the reasons for the changes in the revisit patterns of pediatric patients during the COVID-19 pandemic and whether these changes will be maintained or whether another change will arise after the end of the COVID-19 pandemic.

## Figures and Tables

**Figure 1 children-09-01003-f001:**
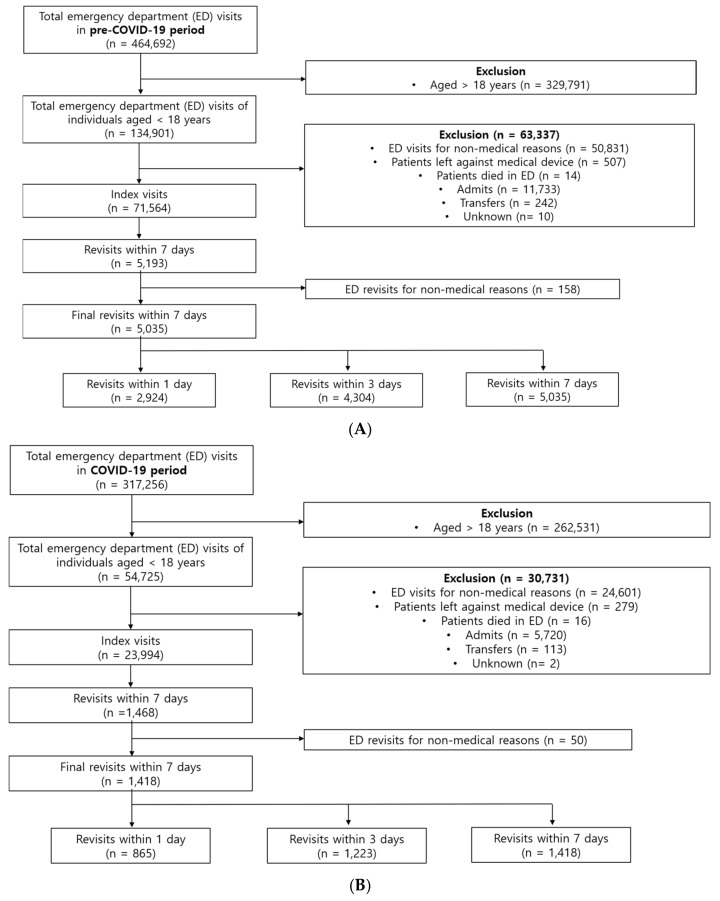
Flow chart of patients enrolled in the pre-COVID-19 period (**A**) and in the COVID-19 period (**B**).

**Figure 2 children-09-01003-f002:**
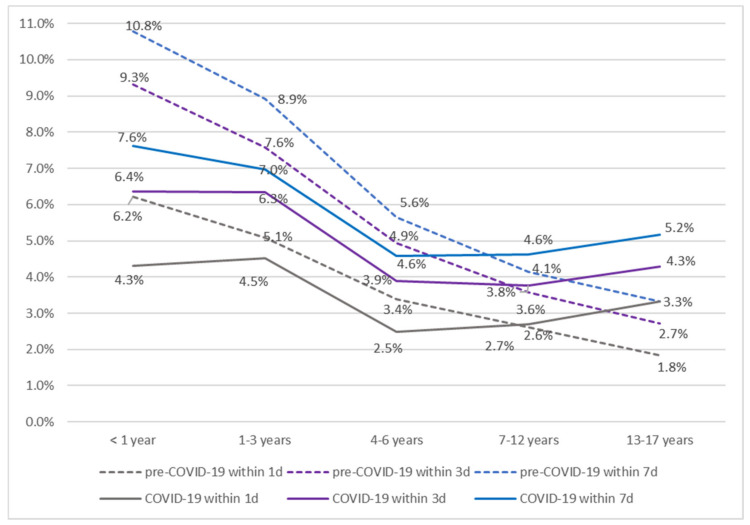
Revisit rates during the pre-COVID-19 and COVID-19 periods based on five age groups.

**Figure 3 children-09-01003-f003:**
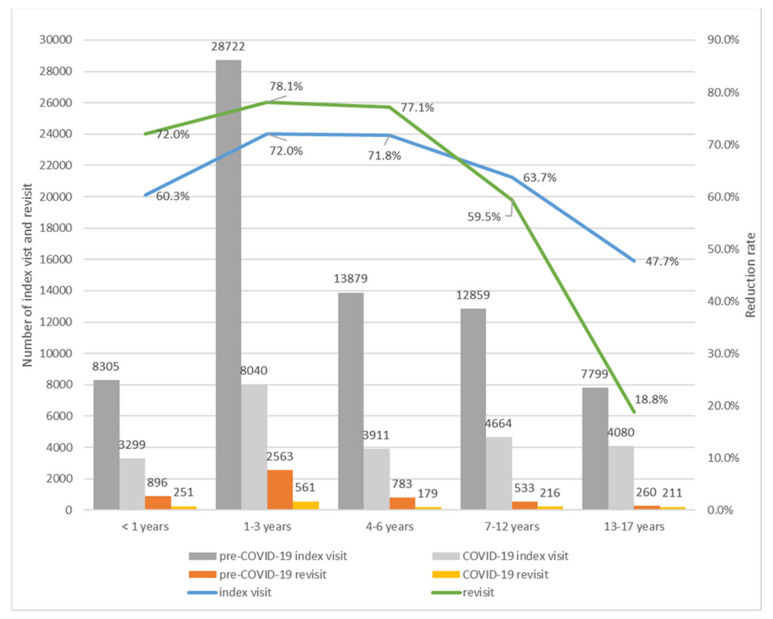
Reduction rates of index visits and revisits.

**Table 1 children-09-01003-t001:** Baseline characteristics of patients at index visits and at 7th-day revisits to the emergency department.

	Pre-COVID-19 Period			COVID-19 Period		
	Index ED Visits *n* = 71,564	Revisits *n* = 5035	*p*-Value	Index ED Visits *n* = 23,994	Revisits *n* = 1418	*p*-Value
Age (years)	4.9 ± 4.7	3.8 ± 3.9	**<0.001**	5.7 ± 5.3	4.9 ± 5.3	**<0.001**
Triage level (KTAS)			**<0.001**			**<0.001**
Level 1	41 (0.1)	3 (0.2)		14 (0.1)	6 (0.3)	
Level 2	2244 (3.1)	9 (0.7)		871 (3.6)	90 (4.8)	
Level 3	40,308 (56.3)	552 (40.2)		12,818 (53.4)	1198 (63.4)	
Level 4	23,468 (32.8)	645 (47.0)		8135 (33.9)	422 (22.3)	
Level 5	5503 (7.7)	163 (11.9)		2156 (9.0)	175 (9.3)	
Transportation to the hospital			0.364			0.295
Self-presentation	68,062 (95.1)	1316 (95.9)		22,372 (93.2)	1775 (93.9)	
Ambulance	3494 (4.9)	56 (4.1)		1622 (6.8)	116 (6.1)	
Vital signs at presentation						
Systolic blood pressure (mmHg)	113.3 ± 14.5	105.7 ± 15.9	**<0.001**	116.7 ± 15.7	114.6 ± 15.0	**0.001**
Diastolic blood pressure (mmHg)	69.1 ± 10.6	65.1 ± 9.8	**<0.001**	71.2 ± 11.8	69.9 ± 11.4	**0.007**
Mean arterial pressure (mmHg)	83.8 ± 10.7	78.7 ± 10.8	**<0.001**	86.4 ± 11.9	84.8 ± 11.5	**0.001**
Pulse rate (times/min)	122.0 ± 26.9	106.5 ± 21.6	**<0.001**	119.4 ± 27.2	123.7 ± 28.4	**<0.001**
Respiratory rate (breaths/min)	24.5 ± 5.1	21.9 ± 3.3	**<0.001**	23.9 ± 5.2	24.7 ± 5.6	**<0.001**
Body temperature (°C)	37.6 ± 1.0	37.3 ± 1.0	**<0.001**	37.2 ± 2.5	37.2 ± 3.9	0.898
ED LOS (minutes)	105 ± 96	113 ± 67	**<0.001**	107 ± 102	169 ± 189	**<0.001**

Categorical data are presented as number (%), and continuous data are presented as mean ± standard deviation; Abbreviations: KTAS, Korea Triage and Acuity Scale; ED, emergency department; LOS, length of stay. Significant values (*p* < 0.05) represented in bold.

**Table 2 children-09-01003-t002:** Top three diagnoses at revisits within 1, 3, and 7 days of the first emergency department visit, stratified by age (%).

Pre-COVID-19 Period						COVID-19 Period				
	<1 Year	1–3 Years	4–6 Years	7–12 Years	13–17 Years	<1 Year	1–3 Years	4–6 Years	7–12 Years	13–17 Years
1 day	Infectious and parasitic diseases (44.9)	Infectious and parasitic diseases (47.1)	Infectious and parasitic diseases (43.9)	Infectious and parasitic diseases (43.8)	Infectious and parasitic diseases (36.4)	Infectious and parasitic diseases (47.5)	Infectious and parasitic diseases (45.8)	Infectious and parasitic diseases (48.5)	Infectious and parasitic diseases (38.9)	Infectious and parasitic diseases (33.8)
	Diseases of the respiratory system (33.9)	Diseases of the respiratory system (29.9)	Diseases of the respiratory system (27.2)	Diseases of the respiratory system (20.2)	Diseases of the digestive system (23.8)	Diseases of the respiratory system (20.6)	Diseases of the respiratory system (24.7)	Diseases of the digestive system (19.6)	Diseases of the digestive system (19.8)	Diseases of the digestive system (19.9)
	Diseases of the genitourinary system (7.4)	Diseases of the genitourinary system (6.0)	Diseases of the digestive system (13.2)	Diseases of the digestive system (17.3)	Diseases of the respiratory system (13.3)	Diseases of the genitourinary system (16.3)	Diseases of the skin and subcutaneous tissue (6.8)	Diseases of the skin and subcutaneous tissue (9.3)	Diseases of the skin and subcutaneous tissue (11.9)	Diseases of the respiratory system (8.1)
3 days	Infectious and parasitic diseases (44.3)	Diseases of the respiratory system (47.1)	Infectious and parasitic diseases (44.5)	Infectious and parasitic diseases (43.9)	Infectious and parasitic diseases (35.1)	Infectious and parasitic diseases (45.7)	Infectious and parasitic diseases (47.1)	Infectious and parasitic diseases (43.4)	Infectious and parasitic diseases (37.5)	Infectious and parasitic diseases (32.0)
	Diseases of the respiratory system (35.5)	Infectious and parasitic diseases (30.8)	Diseases of the respiratory system (28.3)	Diseases of the respiratory system (20.4)	Diseases of the digestive system (23.2)	Diseases of the respiratory system (17.8)	Diseases of the respiratory system (23.8)	Diseases of the digestive system (20.4)	Diseases of the digestive system (22.7)	Diseases of the digestive system (20.0)
	Diseases of the genitourinary system (6.7)	Diseases of the digestive system (5.0)	Diseases of the digestive system (11.4)	Diseases of the digestive system (16.3)	Diseases of the respiratory system (11.8)	Diseases of the genitourinary system (14.9)	Diseases of the skin and subcutaneous tissue (7.6)	Diseases of the respiratory system (10.5)	Diseases of the skin and subcutaneous tissue (10.8)	Diseases of the respiratory system (8.0)
7 days	Infectious and parasitic diseases (43.9)	Diseases of the respiratory system (46.8)	Infectious and parasitic diseases (43.8)	Infectious and parasitic diseases (43.7)	Infectious and parasitic diseases (33.1)	Infectious and parasitic diseases (44.0)	Infectious and parasitic diseases (45.3)	Infectious and parasitic diseases (41.3)	Infectious and parasitic diseases (34.3)	Infectious and parasitic diseases (28.4)
	Diseases of the respiratory system (35.1)	Infectious and parasitic diseases (29.8)	Diseases of the respiratory system (27.6)	Diseases of the respiratory system (20.1)	Diseases of the digestive system (24.6)	Diseases of the respiratory system (17.3)	Diseases of the respiratory system (23.9)	Diseases of the digestive system (19.0)	Diseases of the digestive system (23.6)	Diseases of the digestive system (22.7)
	Diseases of the genitourinary system (6.2)	Diseases of the digestive system (5.3)	Diseases of the digestive system (11.2)	Diseases of the digestive system (15.4)	Diseases of the respiratory system (11.9)	Diseases of the genitourinary system (13.7)	Diseases of the skin and subcutaneous tissue (7.8)	Diseases of the respiratory system (14.5)	Diseases of the skin and subcutaneous tissue (10.2)	Mental and behavioural disorders (7.6)

**Table 3 children-09-01003-t003:** Disposition at the 1st-, 3rd-, and 7th-day emergency department revisits.

		Pre-COVID-19 Period					COVID-19 Period				
	Disposition	<1 Year	1–3 Years	4–6 Years	7–12 Years	13–17 Years	<1 Year	1–3 Years	4–6 Years	7–12 Years	13–17 Years
1 day	Discharge, *n* (%)	301 (58.6)	999 (68.5)	323 (68.6)	231 (68.8)	83 (58.0)	77 (54.6)	234 (64.1)	53 (54.6)	66 (52.4)	79 (58.1)
	General ward, *n* (%)	211 (41.1)	456 (31.3)	147 (31.2)	103 (30.7)	57 (39.9)	63 (44.7)	130 (35.6)	43 (44.3)	59 (46.8)	54 (39.7)
	ICU, *n* (%)	1 (0.2)	1 (0.1)	1 (0.2)	1 (0.3)	1 (0.7)	1 (0.7)	0 (0.0)	0 (0.0)	0 (0.0)	1 (0.7)
	Transfer, *n* (%)	1 (0.2)	2 (0.1)	0 (0.0)	1 (0.3)	2 (1.4)	0 (0.0)	1 (0.3)	1 (1.0)	1 (0.8)	2 (1.5)
	Death, *n* (%)	0 (0.0)	1 (0.1)	0 (0.0)	0 (0.0)	0 (0.0)	0 (0.0)	0 (0.0)	0 (0.0)	0 (0.0)	0 (0.0)
3 days	Discharge, *n* (%)	478 (62.4)	1538 (71.6)	473 (69.0)	316 (68.7)	129 (61.1)	117 (56.3)	345 (67.4)	96 (63.2)	94 (53.4)	112 (64.0)
	General ward, *n* (%)	286 (37.3)	638 (29.3)	212 (30.9)	142 (30.9)	79 (37.4)	88 (42.3)	166 (32.4)	55 (36.2)	80 (45.5)	60 (34.3)
	ICU, *n* (%)	1 (0.1)	1 (0.0)	1 (0.1)	1 (0.2)	1 (0.5)	2 (1.0)	0 (0.0)	0 (0.0)	0 (0.0)	1 (0.6)
	Transfer, *n* (%)	1 (0.1)	2 (0.1)	0 (0.0)	1 (0.2)	2 (0.9)	1 (0.5)	1 (0.2)	1 (0.7)	2 (1.1)	2 (1.1)
	Death, *n* (%)	0 (0.0)	1 (0.0)	0 (0.0)	0 (0.0)	0 (0.0)	0 (0.0)	0 (0.0)	0 (0.0)	0 (0.0)	0 (0.0)
7 days	Discharge, *n* (%)	576 (64.9)	1871 (72.8)	560 (71.4)	371 (69.6)	171 (65.8)	146 (58.9)	386 (68.4)	118 (65.9)	123 (56.9)	136 (64.5)
	General ward, *n* (%)	309 (34.8)	694 (27.0)	223 (28.4)	160 (30.0)	86 (33.1)	99 (39.9)	177 (31.4)	60 (33.5)	91 (42.1)	72 (34.1)
	ICU, *n* (%)	1 (0.1)	1 (0.0)	1 (0.1)	1 (0.2)	1 (0.4)	2 (0.8)	0 (0.0)	0 (0.0)	0 (0.0)	1 (0.5)
	Transfer, *n* (%)	1 (0.1)	3 (0.1)	0 (0.0)	1 (0.2)	2 (0.8)	1 (0.4)	1 (0.2)	1 (0.6)	2 (0.9)	2 (0.9)
	Death, *n* (%)	0 (0.0)	1 (0.0)	0 (0.0)	0 (0.0)	0 (0.0)	0 (0.0)	0 (0.0)	0 (0.0)	0 (0.0)	0 (0.0)

* Abbreviations: ICU, intensive care unit.

## Data Availability

Not applicable.

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
