# Peer review of "A Comparison of Emergency Department Revisit Rates of Pediatric Patients between Pre-COVID-19 and COVID-19 Periods"

_children, 2022, doi:10.3390/children9071003_

Round 1

Reviewer 1 Report

This is an interesting study concerning the characteristics of pediatric patients in the emergency departments (EDs) during the COVID-19 pandemic. Although several authors showed that the EDs’ patients changed during the pandemic, to the best of my knowledge, no studies specifically involved the pediatric population of Korea. 

This is a well-written study. I suggest adjusting table 1 as I did not understand which data the p-value is referred to.

I also suggest citing relevant literature such as DOI: 10.3390/children8060463

Reviewer 2 Report

Thank you for the elaborate documenting of the revisit process during COVID-19 pandemic in children. 

How can we use this observation? What is the take-home message? What do you suggest as potential steps in a future pandemic outbreak?

Please rephrase conclusions.

Also there are some minor revision needed - highlighted in peer-review document 

Reviewer 3 Report

This is a very interesting epidemiological study of ED re-visits in children before and after the COVID-19 epidemic.

#1

Important points for reading the manuscript were mentioned in the final LIMITATION.

・The data were extracted from only the NEDIS data using the registration number of patients, we were unable to confirm whether patients revisited the ED due to the same diagnosis or symptoms of their index visits.

・We could not confirm the purpose of revisit and whether it was scheduled or unscheduled.

These should be properly mentioned within the METHOD.

#2

It would be good to analyze the re-visit rates before and after COVID-19 for each disease category of initial diagnosis.

It would be interesting to see if changes in re-visit rates occurred outside of COVID19-related conditions.

It should be possible to discuss whether the problem is a health care system problem or a disease structure problem.

#3

In TABLES, the legends should be properly described.

The analysis results presented in RESULT should take into account the number of significant digits.

I assume that age, blood pressure (systolic, diastolic, mean), and respiratory rate are written as mean ± SD, but considering the number of significant digits, it is appropriate to write both mean and SD to the first decimal place.

Reviewer 4 Report

An extensive analysis of the national data was carried out on re-visit rates during Covid period. The most important finding is that the 7-17 year age group experienced in increase in re-visit rates. 

Where the schools in the country closed or adopting Web based teaching during this period, making it easier to re-visit ER?

Does reduction of the total ER visit during the pandemic select a group of patients or rather parents who are more concerned about their kids and hence more ready to re-visit ER? 

How would the finding of this research change the practice of a clinician? 

Round 2

Reviewer 4 Report

The study noticed an increased revisit rates of school age children but a decrease for pre-school age children. The possible explanation of the pattern could be explored in relationship of school closure during the pandemic.